# *Ecklonia cava* Polyphenols Have a Preventive Effect on Parkinson’s Disease through the Activation of the Nrf2-ARE Pathway

**DOI:** 10.3390/nu16132076

**Published:** 2024-06-28

**Authors:** Yuri Yasuda, Tamaki Tokumatsu, Chiharu Ueda, Manami Sakai, Yutaro Sasaki, Toshio Norikura, Isao Matsui-Yuasa, Akiko Kojima-Yuasa

**Affiliations:** 1Department of Nutrition, Graduate School of Human Life and Ecology, Osaka Metropolitan University, Osaka 558-8585, Japanyuasa-i@hotmail.co.jp (I.M.-Y.); 2Department of Nutrition, Aomori University of Health and Welfare, Aomori 030-8505, Japan; t_norikura@ms.auhw.ac.jp

**Keywords:** *Ecklonia cava* polyphenols, Parkinson’s disease, Nrf2-ARE pathway, AMPK, SH-SY5Y cells, p62, NQO1, pole test, wire-hanging test

## Abstract

Parkinson’s disease (PD) is a degenerative neurological disorder defined by the deterioration and loss of dopamine-producing neurons in the substantia nigra, leading to a range of motor impairments and non-motor symptoms. The underlying mechanism of this neurodegeneration remains unclear. This research examined the neuroprotective properties of *Ecklonia cava* polyphenols (ECPs) in mitigating neuronal damage induced by rotenone via the activation of the nuclear factor erythroid 2-related factor 2 (Nrf2)–antioxidant response element (ARE) pathway. Using human neuroblastoma SH-SY5Y cells and PD model mice, we found that ECP, rich in the antioxidant polyphenol phlorotannin, boosted the gene expression and functionality of the antioxidant enzyme NAD(P)H quinone oxidoreductase-1. ECP also promoted Nrf2 nuclear translocation and increased p62 expression, suggesting that p62 helps sustain Nrf2 activation via a positive feedback loop. The neuroprotective effect of ECP was significantly reduced by Compound C (CC), an AMP-activated protein kinase (AMPK) inhibitor, which also suppressed Nrf2 nuclear translocation. In PD model mice, ECPs improved motor functions impaired by rotenone, as assessed by the pole test and wire-hanging test, and restored intestinal motor function and colon tissue morphology. Additionally, ECPs increased tyrosine hydroxylase expression in the substantia nigra, indicating a protective effect on dopaminergic neurons. These findings suggest that ECP has a preventative effect on PD.

## 1. Introduction

Parkinson’s disease (PD) is the second most common neurodegenerative disorder globally, after Alzheimer’s disease [1]. Its incidence escalates with age, tripling every decade after 60 [2]. Predictions suggest a drastic rise in PD cases from 6.9 million in 2015 to 14.2 million by 2040, primarily due to the rapid aging of global populations, including Japan [3]. PD manifests with motor symptoms like rigidity, tremor, and motor laxity, along with non-motor symptoms like depression, cognitive impairment, and disturbances in autonomic functions [4]. While therapeutic management can alleviate some symptoms for several years post-onset, PD remains progressive, with no established treatments to halt its course. Consequently, the quest for radical treatments and preventive strategies stands as a principal aim of contemporary PD research [5].

PD comprises two forms: familial, inherited through autosomal dominant or recessive patterns, and sporadic, likely influenced by multiple risk factors. Genetically linked PD constitutes approximately 10% to 15% of cases, while the rest are classified as sporadic [6]. This study focuses on sporadic PD due to its higher incidence. Major risk factors for sporadic PD include genetic and environmental elements. Environmental factors primarily stem from exposure to neurotoxic substances, particularly pesticides detrimental to the human nervous system [7]. While the precise mechanisms underlying dopaminergic neuronal demise remain elusive, various factors, including oxidative stress induced by these agents, mitochondrial dysfunction, the abnormal accumulation of α-synuclein protein, and neuroinflammation, are implicated in PD pathogenesis [8]. Emerging evidence indicates a close association between dopaminergic neuronal death and oxidative stress, notably intracellular reactive oxygen species (ROS) generation [9,10]. Rotenone, sourced from the roots of tropical legumes such as Delis and Couve, serves widely as a pesticide and insecticide. Its mechanism of action involves inducing mitochondrial dysfunction by inhibiting respiratory chain electron transport complex I, consequently elevating ROS production [11]. Heightened ROS levels ultimately trigger apoptosis [12]. Oral rotenone administration has been linked to motor and gastrointestinal dysfunction, alongside a decrease in neurons positive for tyrosine hydroxylase (TH) in the substantia nigra and striatum [13,14]. Thus, rotenone was employed in this study to generate PD models in laboratory settings (*in vitro* and *in vivo*).

The nuclear factor erythroid 2-related factor 2 (Nrf2)–antioxidant response element (ARE) pathway assumes a pivotal role in inducing adaptive responses to oxidative stress [15]. This pathway, a master regulator, activates a cascade of cytoprotective genes, including various transcription factors governing antioxidant enzymes, anti-inflammatory mediators, proteasomes, and mitochondrial biogenesis [16]. Under normal intracellular homeostasis, Nrf2 binds to the cysteine-rich protein Kelch-like ECH-associated protein 1 (Keap1) in the cytoplasm, leading to ubiquitination by Keap1 and subsequent degradation via the ubiquitin–proteasome system [17]. However, the oxidative modification of Keap1’s cysteine residue by electrophiles or ROS induces a conformational change, liberating Nrf2 from Keap1 and ubiquitin, facilitating its translocation into the nucleus [18,19]. Upon entering the nucleus, Nrf2 attaches to ARE, thereby enhancing the expression of antioxidant proteins. Consequently, the antioxidant mechanism of the Nrf2-ARE pathway, an innate mechanism shielding neurons from oxidative stress, garners substantial interest in PD prevention and treatment [20].

In the realm of oxidative stress, natural antioxidants like polyphenols play a crucial role. Polyphenols function as antioxidants either directly by scavenging or preventing ROS or indirectly by enhancing endogenous antioxidant defense mechanisms. Abundant in various foods and beverages, polyphenols exhibit diverse biological activities, encompassing antioxidative, free radical-scavenging, metal ion-chelating, vasoprotective, hepatoprotective, anticancer, anti-infective, and anti-inflammatory properties [21]. Hence, the avenue of PD prevention via dietary components assumes significance. *Ecklonia cava* (known as “kajime” in Japanese) stands as an edible brown alga thriving in warm coastal regions of the Pacific Ocean, classified under the kelp family *Kelpidae* [22]. Rich in phlorotannin, a polyphenol unique to brown algae, *Ecklonia cava* provides a range of biological activities, such as antioxidant, antibacterial, and anti-inflammatory effects [23]. Notably, our prior research indicates that *Ecklonia cava* polyphenol (ECP) treatment counteracts ethanol-induced hepatocyte mortality by neutralizing ROS and via a cyclic AMP-dependent mechanism [24,25].

This study aims to elucidate the relationship between ECP’s neuroprotective effect against rotenone-induced neuronal damage and its antioxidant mechanism through the Nrf2-ARE pathway in an *in vitro* experimental setup. Additionally, we aim to delineate ECP’s protective efficacy against rotenone-induced PD via *in vivo* experiments employing rotenone-induced PD model mice.

## 2. Materials and Methods

### 2.1. ECP Formulation

We utilized a commercially available polyphenol extract from *Ecklonia cava* (Seapolynol, Livechem Inc., Jeju, Korea). The *Ecklonia cava* extract had a total polyphenol content of 99.4%, determined using Folin–Ciocalteu reagent with phloroglucinol as a standard. High-performance liquid chromatography (HPLC) analysis identified several significant phlorotannins in the extract, including dieckol (8.2%), 8,8′-bieckol (2.8%), 2-O-(2,4,6-trihydroxyphenyl)-6,6′-bieckol (2.1%), 6,6′-bieckol (1.5%), phlorofurofucoeckol-A (1.4%), eckol (0.6%), 2-phloroeckol (0.4%), 7-phloroeckol (0.4%), and phlorotannin A (0.4%). The HPLC analysis was performed using a CAPCELL PAK ODS column (4.6 × 250 mm, Waters) with an elution solvent of 30% aqueous MeOH at a flow rate of 0.8 mL/min [26].

### 2.2. Culture of Human Neuroblastoma Cells (SH-SY5Y)

Human neuroblastoma (SH-SY5Y; KAC Co., Ltd., Kyoto, Japan), the most commonly used cell line in PD-related studies, was maintained in Dulbecco’s Modified Eagle’s Medium (DMEM) supplemented with 10% fetal bovine serum (FBS) at 37 °C in a humidified atmosphere with 5% CO_2_. The medium was refreshed every 2–3 days, and cells were subcultured when reaching 70-80% confluence. ECP was dissolved in dimethyl sulfoxide (DMSO; FUJIFILM Wako Pure Chemical Corporation, Osaka, Japan) and used in the experiments.

### 2.3. Cell Viability

Cell viability was assessed using the MTT assay with 3-(4,5-Dimethyl-2-Thiazolyl)-2,5-Diphenyltetrazolium Bromide. SH-SY5Y cells (2.0 × 10^4^ cells) were cultured overnight in 96-well plates to facilitate attachment. The cells were treated with varying concentrations of ECP (0–50 µg/mL) and rotenone (0–400 nM; Sigma-Aldrich Japan K.K., Tokyo, Japan). Afterward, the medium was aspirated, and each well was incubated with 100 µL of medium containing 10% MTT solution (5 mg/mL) for 2–4 h. Subsequently, 200 µL of DMSO was added to each well. The plates were then agitated using a plate mixer (BIOTEC Co., Ltd., Tokyo, Japan), and the absorbance at 535 nm was measured using a microplate reader (Wallac 1420 ARVOsx, PerkinElmer, Inc. Waltham, MA, USA).

### 2.4. Assay of Intracellular ROS Level

The measurement of intracellular reactive oxygen species (ROS) production was conducted using 2′,7′-Dichlorofluorescin diacetate (DCFH-DA), a probe that permeates cell membranes and is particularly sensitive to hydrogen peroxide. SH-SY5Y cells were treated with 2.4 mM DCFH-DA for the last 30 min of a 6 h incubation period. After incubation, cells were washed twice with PBS (-), and intracellular fluorescence was visualized using FSX100 Bio Imaging Navigator (Olympus Corporation, Tokyo, Japan). ROS levels were quantified by measuring fluorescence intensity using ImageJ software 1.52 Version.

### 2.5. Gene Expression Levels

A quantitative reverse transcription PCR (qRT-PCR) was utilized to assess gene expression levels. Total RNA was extracted from SH-SY5Y cells using the High Pure RNA Isolation Kit (Roche Diagnostics GmbH, Mannheim, Germany) following the manufacturer’s instructions. RNA quality was evaluated with the Agilent RNA 6000 Nano Kit (Agilent Technologies, Santa Clara, CA, USA). Complementary DNA (cDNA) was synthesized from the extracted RNA using the ReverTra Ace^®^ qPCR RT Master Mix (Toyobo Co., Ltd., Osaka, Japan). The qRT-PCR was conducted with the GeneAce SYBR™ qPCR Mix (Nippon Gene Co., Ltd., Tokyo, Japan) on a Stratagene Mx 3005P instrument (Agilent Technologies). Gene expression levels were normalized to the housekeeping gene β-actin, and the fold change relative to the control group was determined using the ΔΔCt method. The mRNA expression levels were derived from this calculation. Primers were designed using Primer-BLAST (http://www.ncbi.nlm.nih.gov/tools/primer-blast/, accessed on 6 September 2022), and the specificity of the amplified products was verified by melting curve analysis. Primer sequences are provided in Table 1.

### 2.6. NQO1 Activity

NQO1 is an antioxidant enzyme that converts toxic quinones, which contribute to ROS production in cells, into hydroquinones. The activity of NQO1 was determined by measuring the dicoumarol-inhibitable reduction of DCPIP. In this experiment, SH-SY5Y cells (2.0 × 10^5^ cells/mL in a 35 mm dish) were treated with ECP (12.5 µg/mL) and rotenone (200 nM) for 6 h. Following treatment, the cells were collected, resuspended in ice-cold TE buffer, and subjected to three freeze–thaw cycles using liquid nitrogen and a 37 °C water bath. The cell lysates were then centrifuged at 12,000× *g* for 5 min at 4 °C. Protein concentrations in the supernatant were measured using the Pierce BCA Protein Assay Kit (Thermo Fisher Scientific, Inc., Waltham, MA, USA). To assess NQO1 activity, a reaction mixture with a final volume of 1 mL was prepared, containing 50 µL of the supernatant. The reaction was started by adding 2 µL of 40 µM 2,6-Dichloroindophenol Sodium Salt (DCPIP; FUJIFILM Wako Pure Chemical Corporation, Osaka, Japan) additionally, with or without 5 µL of 20 µM dicoumarol (FUJIFILM Wako Pure Chemical Corporation). The reduction of DCPIP was monitored at 28 °C for 1 min at 600 nm (ε = 21 × 10^3^ M^−1^cm^−1^).

### 2.7. Nuclear Nrf2 Expression: Immunofluorescence Staining

Cells were cultured overnight in a Lab-Tek Chamber Slide System (Thermo Fisher Scientific, Inc.) and treated with ECP (12.5 µg/mL) and rotenone (200 nM) for 3 h. After treatment, the cells were washed twice with 1 mL of PBS, fixed with MeOH (FUJIFILM Wako Pure Chemical Corporation) for 5 min at −20 °C, air-dried, and then permeabilized with 0.1% Triton-X (FUJIFILM Wako Pure Chemical Corporation) for 10 min. Following this, the cells were blocked with 2 drops of Protein Block Serum-free (X0909, Agilent Technologies, Inc., Santa Clara, CA, USA) for 30 min. The primary antibody against Nrf2 (sc-365949, Santa Cruz Biotechnology, Inc., Dallas, TX, USA), diluted 1:40 with PBS, was applied and incubated overnight at 4 °C. After two PBS washes, cells were incubated with Alexa Fluor^TM^ 488-conjugated goat anti-mouse IgG (A-11001, Life Technologies, Carlsbad, CA, USA) diluted 1:200 with PBS for 1 h in light-shielded conditions. Finally, cells were stained with 4′,6-diamidino-2-phenylindole dihydrochloride (DAPI, FUJIFILM Wako Pure Chemical Corporation) for 5 min to visualize the nuclei. The nuclear translocation of Nrf2 was observed using an FSX100 Bio Imaging Navigator (Olympus Corporation). A quantification of Nrf2 fluorescence intensity in the nuclei was analyzed using ImageJ software.

### 2.8. Animal Treatment

Animal experiments were conducted following the Regulations on Animal Experiments at Osaka City University (now Osaka metropolitan University) and were approved by the Experimental Animal Ethics Committee (permission number: S0073). Twenty-three 6-week-old male C57BL6/J mice weighing 20–22 g were procured from SLC, Inc. of Japan (Shizuoka, Japan). All the mice were housed at a constant temperature (25 °C) under a 12 h light/dark cycle (8:00 am to 8:00 pm light). They had ad libitum access to a control diet (see Table 2) and water.

After 6 days of pre-rearing to acclimate the mice to the laboratory conditions, they were randomly assigned to four groups (6 mice for each group) as follows:Control group;Rotenone group;ECP(L) group—treated with a low concentration of ECP (20 mg/kg body weight);ECP(H) group—treated with a high concentration of ECP (240 mg/kg body weight).

Groups 2 to 4 received an oral administration of rotenone (10 mg/kg body weight) daily for 30 days, beginning 1 week after the start of the study. Rotenone was dissolved in a solution containing 3% carboxymethyl cellulose sodium salt (CMC, FUJIFILM Wako Pure Chemical Corporation) and 1.25% chloroform (FUJIFILM Wako Pure Chemical Corporation). The control group received only the solvent.

### 2.9. Motor Function Test

#### 2.9.1. Pole Test

The pole test, conducted once a week for five weeks, aimed to assess the degree of slow movement, a prominent motor symptom of PD (Figure 1). Mice were positioned with their heads facing upwards atop a 75 cm-high pole perpendicular to the floor. The T-turn, defined as the time taken for the mouse to change direction from the initial position to a downward head-down posture, was recorded. Additionally, the time taken for the mouse to complete the T-turn and reach the ground (total time) was measured three times per day. A prolonged T-turn and total time indicated the onset of motor dysfunction.

#### 2.9.2. Wire-Hang Test

Due to the observed weekly recovery of motor function with ECPs in the pole test, another motor assessment, the wire-hang test, was conducted on day 25 following rotenone administration. This test aimed to evaluate muscle strength. Mice were permitted to cling to a wire mesh, which was then inverted, and the duration until the mouse fell from a height of 35 cm was recorded for up to 5 min. A shorter time to fall indicated the presence of motor dysfunction.

### 2.10. Intestinal Motility: Evans Blue Dye Administration

PD patients often experience non-motor symptoms, including autonomic and psychiatric symptoms, in addition to motor symptoms [14]. Consequently, assessing gastrointestinal motility dysfunction, a common autonomic nervous system complication observed in PD, is crucial. The Evans blue dye migration distance from the pylorus was measured to evaluate intestinal motility function [14]. Twenty minutes prior to dissection, each mouse received 0.3 mL of a 2.5% Evans blue solution orally. Following dissection, the distance traveled by the blue dye, measured from the pylorus to the point furthest reached by the dye, was recorded. The Evans blue solution consisted of CMC, chloroform (FUJIFILM Wako Pure Chemical Corporation), and Evans blue dye (NAKALAI TESQUE, Inc., Kyoto, Japan).

### 2.11. Histopathological Analysis of Colon Tissue: Hematoxylin and Eosin (HE) Staining

Colon tissues were fixed in a 10% neutral buffered formalin solution. The fixed intestinal samples were sent to the Nutrition and Pathology Laboratory, Inc., for histopathological specimen preparation. The sections stained with HE staining were examined under a microscope and photographed in their entirety.

### 2.12. Tyrosine Hydroxylase (TH) Staining in Brain Tissue

Brain tissues were fixed in a 10% neutral buffered formalin solution. The fixed brain samples were sent to the Biopathology Institute, Inc. for immunohistological specimen preparation. TH is the rate-limiting enzyme in dopamine synthesis, catalyzing the conversion of tyrosine to L-dopa. TH expression serves as a biomarker for dopaminergic neurons in the substantia nigra of the midbrain. TH expression was visualized using the LSAB method with an anti-TH antibody (AB152, Sigma-Aldrich Co. LLC, St. Louis, MO, USA) diluted 1:80 with PBS. After preincubation with 3% hydrogen peroxide to block endogenous peroxidase for 5 min and subsequently with Protein Block Serum-free (X0909, Agilent Technologies) to inhibit non-specific reactions for 5 min, the specimens were incubated with primary antibody for 1 h at room temperature. They were incubated with polyclonal goat anti-rabbit immunoglobulins/Biotinylated (E0432, Agilent Technologies) diluted 1:400 with PBS for 30 min and then incubated with Streptavidin/HRP (P0397, Agilent Technologies) diluted 1:400 with PBS for 30 min. They were rinsed in PBS after each incubation step. The peroxidase reaction was visualized using 3,3′-diaminobenzidine tetrahydrochloride (DOJINDO LABORATORIES, Kumamoto, Janpan) with nickel chloride color modification. The expression of TH was observed under a microscope (IX-70, Olympus Corporation). The quantification of TH staining intensity was analyzed using ImageJ software.

### 2.13. Statistical Analysis

All quantitative data were presented as mean ± standard deviation (SD) or mean ± standard error (SE). Group differences were assessed using one-way ANOVA. For multiple comparisons, the Tukey–Kramer method was employed, and significance differences were determined at either a 5% or 1% risk rate.

## 3. Results

### 3.1. Effects of Rotenone and ECP on SH-SY5Y Cell Viability

The effect of rotenone and ECP on the viability of SH-SY5Y cells at 24 h was evaluated using the MTT method. ECP had no effect on cell viability at any concentration from 0 to 50 µg/mL (Figure 1A). To determine the optimal rotenone concentration for the *in vitro* PD model, cells were treated with various concentrations (0–400 nM) of rotenone. Cell viability was significantly reduced in a dose-dependent manner (Figure 1B). Therefore, in this study, 200 nM rotenone was used in subsequent experiments as a model of neuronal damage. The protective effect of ECP against rotenone-induced cell injury was evaluated. The results showed that cell viability, which was significantly reduced by the addition of 200 nM rotenone, was significantly restored to control levels with 12.5 µg/mL or 25 µg/mL ECP (Figure 1C). Based on these results, 12.5 µg/mL ECP was used in subsequent experiments.

### 3.2. Effects of Rotenone and ECP on Intracellular ROS Production

Rotenone is known to enhance the production of ROS, leading to oxidative stress. To visualize ROS, cells were stained with the membrane-permeable probe DCFH-DA, which fluoresces upon oxidation. Cells treated with rotenone alone exhibited a significant increase in green fluorescence after 6 h of incubation (Figure 2A,B). In contrast, cells simultaneously treated with ECP showed fluorescence levels comparable to the control group. These findings indicate that ECP effectively inhibits rotenone-induced ROS production.

### 3.3. Effect of ECP on the Gene Expression Level and Activity of NQO1, an Antioxidant Enzyme

To elucidate the mechanism by which ECP suppresses ROS production, we investigated the gene expression level and activity of the antioxidant enzyme NQO1. NQO1 hydroxylates toxic quinones, which are involved in ROS production, thereby contributing to cellular protection against oxidative stress [27]. Our results demonstrated that both the gene expression level and the enzymatic activity of NQO1 were significantly elevated in the ECP-treated group compared to the group treated with rotenone alone after 6 h of incubation (Figure 3A,B).

### 3.4. Effects of Rotenone and ECP on Nuclear Translocation of the Transcription Factor Nrf2

The nuclear translocation of Nrf2 is crucial for the activation of the Nrf2-ARE pathway, which is a key regulator of the oxidative stress response. This process promotes the expression of antioxidant enzymes, such as NQO1, by facilitating the binding of Nrf2 to AREs on DNA. To assess the effects of rotenone and ECP on Nrf2 nuclear translocation, we conducted immunofluorescence staining. The control group exhibited minimal nuclear Nrf2 expression, whereas the rotenone-treated group showed a slight increase in nuclear Nrf2. Notably, the group treated with both ECP and rotenone displayed a more pronounced nuclear translocation of Nrf2 compared to the rotenone-alone group (Figure 4A,B). These results suggest that ECP enhances the nuclear translocation of Nrf2.

### 3.5. Effects of Rotenone and ECP on p62 mRNA Expression Levels, a Nrf2 Target Gene

Given the observed promotion of the nuclear translocation of Nrf2 by ECP, we postulated its involvement in Nrf2 activation and investigated the expression level of p62 mRNA, a target gene of Nrf2. The induction of the p62 gene is mediated by Nrf2 [28], and it has been reported that the p62 protein directly interacts with Keap1, thereby contributing to Nrf2 activation. Moreover, the p62 protein facilitates sustained Nrf2 activation through a positive feedback loop in the Nrf2 pathway [28,29]. Following 4 h of incubation, we observed a significant increase in p62 mRNA expression with the addition of ECP compared to the rotenone-alone group (Figure 5). This finding supports the notion that ECP promotes the nuclear translocation of Nrf2, indicating its involvement in Nrf2 activation.

### 3.6. Effect of Compound C (CC), an AMPK Inhibitor, on SH-SY5Y Cell Viability

Given the role of AMPK in regulating pathways involved in the pathogenesis of various PD factors and hypothesizing that AMPK activation contributes to the protective effect of ECP, we evaluated cell viability with the addition of CC, an AMPK inhibitor. The results demonstrated that ECP significantly restored cell viability; however, this protective effect was notably suppressed when CC was administered 30 min prior to the main culture [30] (Figure 6). These findings suggest that AMPK activation is integral to the protective effect of ECP.

### 3.7. Effects of Rotenone and ECP on Nrf2 Nuclear Migration by AMPK Activation

Immunofluorescence staining was conducted to assess the impact of CC addition on the ECP-induced nuclear translocation of Nrf2. The results indicated a significant suppression of Nrf2 nuclear translocation by CC in the presence of ECP (Figure 7A,B). In contrast, minimal nuclear Nrf2 expression was observed in the control and CC-alone groups, while the rotenone-alone group showed only a slight expression. Moreover, ECP administration markedly promoted the nuclear translocation of Nrf2, a response that was attenuated by CC addition. This suggests the involvement of AMPK activation in mediating the nuclear migration of Nrf2 induced by ECP.

### 3.8. Effects of Rotenone and ECP on Motor Function in Mice

The locomotor activity of mice was evaluated using the pole test. Weekly assessments revealed a progressive prolongation in the T-turn time in the rotenone-treated group until day 26 of treatment, whereas the ECP-treated group exhibited significantly shorter T-turn times. Notably, on days 12 and 19, the T-turn time was notably shorter in the ECP group compared to the rotenone group (Figure 8A). Additionally, the total time to complete the task was significantly reduced in the ECP group compared to the rotenone group, which was particularly evident on day 12, where the ECP (H) group exhibited T-turn times comparable to the control group (Figure 8B). Furthermore, a wire-hang test, assessing muscle strength, was conducted on day 25 of treatment. The results indicated a trend towards a shorter time to fall in the rotenone group and an extended time to fall in the ECP group (Figure 8C).

### 3.9. Effects of Rotenone and ECP on Intestinal Motor Function in Mice

In addition to motor symptoms, PD patients often experience nonmotor symptoms, including autonomic and psychiatric symptoms [14]. Therefore, we examined gastrointestinal motor dysfunction, a common autonomic complication in PD. To assess intestinal motor function, we measured the migration distance of Evans blue dye from the pylorus. The distance traveled by the rotenone-treated group exhibited a decreasing trend compared to the control group, indicative of impaired intestinal motor function. However, the administration of ECP showed a trend towards the restoration of intestinal motor function (Figure 9).

### 3.10. Effects of Rotenone and ECP on the Morphology of Mouse Colon Mucosa

The morphology of the colon mucosa was assessed through HE staining. In comparison to the control group, the rotenone-treated group displayed shortened intestinal mucosal layers and exhibited signs of atrophy or loss of intestinal crypts. Conversely, the ECP-treated group showed minimal abnormalities in the colon mucosa, maintaining a morphology similar to that of the control group (Figure 10).

### 3.11. Effects of Rotenone and ECP on TH Expression in Midbrain Substantia Nigra of Mice

To assess neurodegeneration associated with motor deficits, the immunohistochemical staining of TH in the substantia nigra was conducted. In comparison to the control group, the rotenone-treated group significantly exhibited a reduction in TH-positive dopaminergic neurons, indicative of neurodegeneration. Conversely, a significant recovery of TH-positive expression was observed in the ECP-treated group, and the ECP(H) group showed TH expression levels close to those of the control group (Figure 11A,B).

## 4. Discussion

*In vitro* experimental systems have demonstrated that ECP demonstrates protective effects against rotenone-induced neuronal damage, primarily through activation of the Nrf2-ARE pathway. ECP treatment promoted the expression of the antioxidant enzyme NQO1 and the gene p62, which is crucial for the sustained activation of Nrf2. Additionally, the use of AMPK inhibitors revealed that ECP enhances cell viability and promotes the nuclear translocation of Nrf2 through AMPK activation, suggesting that AMPK acts as an upstream regulator in this protective mechanism.

Firstly, our study established that ECP is not cytotoxic at concentrations up to 50 µg/mL. Conversely, rotenone, widely used to model PD, exhibited cytotoxicity in SH-SY5Y cells, significantly reducing cell viability in a manner dependent on its concentration, particularly at concentrations exceeding 200 nM. Consequently, 200 nM rotenone was selected for generating the *in vitro* PD model in this study. To determine the effective concentration of ECP against rotenone-induced neuronal damage, ECP and rotenone were added simultaneously, and cell viability was assessed. The results indicated that 12.5 µg/mL of ECP restored cell viability, which was significantly reduced by rotenone, to control levels. These findings suggest that low doses of rotenone (200 nM) induce neuronal damage, and 12.5 µg/mL of ECP provides a protective effect against this damage.

The mesencephalic substantia nigra contains high levels of oxidizable substances such as dopamine, and exposure to pesticides like rotenone leads to excessive ROS production. This sustained oxidative stress induces the death of dopaminergic neurons, thereby contributing to the development of PD [11,12]. Therefore, the inhibition of ROS production is critical for preventing PD. In this study, ECP addition significantly suppressed intracellular ROS production, which was markedly enhanced by rotenone after 6 h of culture, restoring it to control levels. Consequently, we investigated the mechanism by which ECP suppresses ROS production.

These results underscore the potential of ECP in mitigating neuronal damage through AMPK activation and ROS suppression, highlighting its therapeutic promise for conditions such as PD. Further investigations are needed to clarify the precise mechanisms of ECP action and its potential applications in neuroprotection.

The Nrf2-ARE pathway plays a crucial role as an endogenous antioxidant mechanism. The transcription factor Nrf2 plays a central role by interacting with the ARE and activating gene transcription, either constitutively or in response to oxidative stress signals [31]. Upon activation, Nrf2 moves into the nucleus and binds to the ARE, leading to the activation of genes that encode antioxidant enzymes [32]. Consequently, Nrf2 is being explored as a therapeutic target for neurodegenerative diseases. The antioxidant enzyme NQO1, which is upregulated by Nrf2, catalyzes the reduction of quinones and may mitigate oxidative stress resulting from the oxidation of dopamine to dopamine quinone [33].

In this study, ECP markedly promoted the nuclear translocation of Nrf2 compared to rotenone alone and markedly increased NQO1 mRNA expression and NQO1 activity. ECP also significantly upregulated the expression of p62, a known target gene of Nrf2. Previous research has shown that oxidative stress-induced upregulation of the p62 gene involves Nrf2, and the p62 protein itself contributes to the activation of Nrf2 [29]. These findings suggest that ECP may bolster the antioxidant response through the activation of Nrf2 (Figure 2).

However, further research is needed to elucidate this mechanism fully, specifically by measuring the protein levels of p62 following ECP treatment. This will provide a clearer understanding of how ECP enhances the Nrf2-mediated antioxidant response and its potential therapeutic applications in neurodegenerative diseases.

AMPK is a serine/threonine kinase that senses energy levels and is widely expressed in various tissues. Its activation enables cells to restore energy homeostasis [34]. AMPK has been shown to cause Nrf2 accumulation in the nucleus through the phosphorylation of Nrf2 [35,36]. AMPK has been recognized as a novel kinase that promotes Nrf2 activation via nuclear accumulation, highlighting its unique role in enhancing antioxidant capacity and cell survival [37]. It is reported that dieckol, a phlorotannin extracted from *Ecklonia cava*, suppresses adipogenesis by activating AMPK in 3T3-L1 preadipocytes [38]. Furthermore, Leem et al. have reported that creatine supplementation with exercise reduced α-synuclein oligomerization by activating AMPK [39]. In this study, the pre-addition of Compound C, an inhibitor of AMPK [30], significantly suppressed the protective effect of ECP against neuronal damage and markedly inhibited the nuclear translocation of Nrf2 promoted by ECP. These results suggest that the protective effect of ECP against neuronal damage involves the AMPK-mediated activation of Nrf2. Additional research is required to clarify the mechanisms by which ECP activates AMPK and to explore the interaction between AMPK and Nrf2.

In this *in vivo* experimental study, a PD model was created by administering rotenone (10 mg/kg body weight) orally to mice daily for 30 days. Oral administration of thalidomide has previously been reported to cause motor and gastrointestinal dysfunction, along with a decrease in TH-positive dopamine neurons in the substantia nigra and striatum [13,14]. Here, motor function was assessed using the pole test to measure movement slowness, and the wire-hang test to evaluate grip and muscle strength. Importantly, various nonmotor symptoms of PD frequently appear before the onset of motor symptoms. Recent research has underscored the role of the enteric nervous system (ENS) in the gastrointestinal symptoms linked to PD pathogenesis [40,41,42]. Among these nonmotor symptoms, gastrointestinal dysfunction is the most common, with delayed gastric emptying frequently reported in PD patients [43]. Gastrointestinal motility was evaluated by measuring the distance Evans blue dye traveled from the pylorus following oral administration [14]. Additionally, the morphology of the colonic mucosa was observed. Given that PD pathology is marked by the loss of dopaminergic neurons in the substantia nigra, brain tissue was analyzed for TH expression, which serves as a marker for these neurons. Our results demonstrated that the oral administration of rotenone led to both motor and gastrointestinal dysfunction, which were subsequently reversed by the oral administration of ECP. ECP treatment maintained normal colonic tissue morphology and increased the population of dopaminergic neurons in the substantia nigra. The observed increase in TH-positive dopaminergic neurons supports the conclusion that ECP administration restored motor function. A critical challenge with brain-targeted therapeutics is their ability to cross the blood–brain barrier (BBB). Kwan and colleagues report that dieckol labeled with fluorescein isothiocyanate and rhodamine B is detected in the cortex and hippocampus within 20 min after injection. Furthermore, they have clarified that this uptake through the BBB occurs via membrane diffusion rather than by phlorotannin membrane transporters [44]. Additionally, many studies have reported that phlorotannins have effects on cognitive processes such as learning and memory [45,46]. This suggests that the protective effect of ECP on brain tissue may be attributable to this property. These results indicate that ECP has a protective effect in a PD animal model. However, it is necessary to further investigate which phlorotannin within ECP is responsible for this effect.

## 5. Conclusions

Our study demonstrates that ECP exerts neuroprotective effects via the Nrf2-ARE pathway in an *in vitro* experimental system and offers protection against PD in an *in vivo* model. Further studies are needed to elucidate the detailed mechanisms of ECP action in both *in vitro* and *in vivo* systems to facilitate its practical application.

## Data Availability

The original contributions presented in the study are included in the article material; further inquiries can be directed to the corresponding author.

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
