# Peer review of "Ecklonia cava Polyphenols Have a Preventive Effect on Parkinson’s Disease through the Activation of the Nrf2-ARE Pathway"

_nutrients, 2024, doi:10.3390/nu16132076_

Round 1

Reviewer 1 Report

Comments and Suggestions for Authors

Dear Authors,

the manuscript presented accurately displays the neuroprotective effect of Ecklonia cava’s polyphenol on early neurodegeneration, histopathological alterations and motor symptoms in a valuable model for Parkinson’s disease. The manuscript is, given its novelty, topic and results. The paper is well-written and well-presented; however, a few aspects of the manuscript need revision from the authors.

Briefly,

·       > In the material and methods section when an antibody is mentioned, the provider company, the catalog number associated with the specific product should be specified alongside the dilution factor of the solution employed in the assay (line 174; 174; 254)

·       >Tyrosine Hydroxylase is correctly stated as limiting factor in dopamine synthesis, however its function is limited to the synthesis of L-dopa, while another enzyme (AADC) is responsible for the synthesis of dopamine from L-dopa. I suggest simply replacing “dopa” with “L-dopa” at line 252

·      > All assays are extensively described in the material and methods section; however, the immunohistochemical assay at section 2.12 is lacking in detail regarding the protocol employed, I suggest including more details regarding sample preparation and the immunohistochemical staining procedure (e.g. antibody dilution, time, temperature of the antibody reaction, solution used for washings etc).

·      > In section 3.1 of the results at line 268 the authors describe the experimental setting as “a model of nerve injury” which in my opinion would be incorrect. I suggest replacing “nerve injury” with “neuronal damage” a sentence later correctly employed by the authors at line 734 and 748

·       >(I am guessing here) there has been a minor font mistake in the Figure1 caption which should be addressed.

·      > Figure10 and figure11 properly introduce the images shown, but an explanation of the results displayed should be reserved for the results section. Therefore, I suggest removing the comments at lines 694-696 and 729-730.

·      > Figure10 and figure11 report incorrect magnifications. The magnification in a light microscope comes from the product of the lens magnification (4X for figure10 and 10 for figure11) with the eyepiece magnification (10X). Please replace the magnifications considering this eyepiece component.

·      > At lines 244-245 the authors imply to have performed an hematoxylin and eosin staining on the brain samples while no images are then submitted with the manuscript. I suggest either providing those images since they could be of interest for the study (specially for the substantia nigra region) or rephrasing paragraph 2.11 to eliminate this possible misunderstanding

Although I raised numerous points to be corrected or revised, I firmly believe the manuscript submitted is of great value and the variety and completeness of the experiment performed make it a great work. Moreover, I appreciate many aspects of the introduction as it correctly present the topic and the scientific design, and of the discussion, which is a proper interpretation of the data presented and acknowledges the importance of p62 protein concentration for further studies.

Reviewer 2 Report

Comments and Suggestions for Authors

Paper by Yasuda et al., investigates on the neuroprotective effects of an Ecklonia cava extract in in vitro and in vivo models of Parkinson’s disease. Paper is well written and endowed by scientific soundness. However, it needs to be improved. This is because the manuscript is lacking in some points.

1)        Authors should be clearer in the title, since they tested a polyphenolic extract of Ecklonia cava and not a single polyphenol, as revealed by HPLC analysis.

2)        Authors should be more consistent. This is because they reported in the abstract that ECP is rich in the antioxidant polyphenol phlorotannin, whereas the HPLC analysis revealed that phlorotannin accounts only for the 0.4% of whole extract.

3)        Authors should be more precise in the description of their experimental protocol. In Scheme 1 it appears that ECP was administered for 7 days, then followed by the addition of rotenone for further 30 days. However, Authors refer to a simultaneous administration of ECP and rotenone throughout their paper. Therefore, Authors should clarify this aspect. Furthermore, the pre-administration of some compound/extracts may produce a protective effect against a certain damage. On this basis, in order to have a protective effect, each compound/extract should be administered before the stressor (i.e., rotenone). How do Authors justify this aspect?

4)        Authors must specify the number of mice used for each group. This number must be the same for each group.

5)        Authors should indicate why they did not consider the experimental group of mice treated only with ECP L/H.

6)        NRF2 activation is associated with the defense response against oxidative damage. Why did this activation occur in the rotenone-treated group? The data in Figure 4 are insufficiently explained. Similar trends occur for results on p62 gene expression. Please, Authors should better justify and explain the interpretation of these results.

7)        Figure 6 is confusing and unclear. Please replace it with another one. The third and fifth bars of the histogram of Figure 6 show a same type of treatment and different results.

8)        Authors must add statistical analysis in Figure 8 and 9.

9)        Authors should add the scale in Figures 10 and 11.

10)    Authors should add a graph showing the NRF2-ARE pathway.

Comments on the Quality of English Language

English is fine. Minor editing are required.

Reviewer 3 Report

Comments and Suggestions for Authors

The authors of this work used PD model mice and rotenone-induced models of neuronal damage in human neuroblastoma SH-SY5Y cells to examine the protective potential of Ecklonia cava polyphenol (ECP). Phlorotannin-rich ECP markedly elevated NQO1 antioxidant enzyme expression and activity as well as Nrf2 nuclear translocation, indicating a p62-mediated positive feedback loop that maintains Nrf2 activation. In PD model mice, ECP restored intestinal morphology and enhanced motor functions in vivo. The protective effects were lessened by Compound C, an AMPK inhibitor, suggesting that ECP's action depends on AMPK activation. Overall, due to its antioxidant qualities and ability to modulate the Nrf2-ARE pathway, ECP showed promise as a preventive treatment for Parkinson's disease. While there has been some interest in the authors' research, some of the key pathological features of PD such as α-synuclein aggregation, mitochondrial dysfunction, impaired proteostasis, and loss of dopaminergic neurons have not been examined. This significantly lessens the study's relevance to PD pathology. In addition, the research design and results analysis contain a number of ambiguities, some of which are explained below. 

1.     Figures 1A and 1C demonstrate that rotenone has an activity of roughly 90% and less than 80% at 200 nM, respectively. Furthermore, numerous individuals have carried out analogous studies wherein rotenone concentrations were typically within the μM range (e.g., Fig. 1 in Behav. Brain Funct. 2013;9:13); in contrast, the authors solely employed sub-μM concentrations, which deviate significantly from the widely accepted IC50 conditions. Why? 

2.     Figure 2: In a bright field, each group's cell count was constant and similar. How then did the authors look into the possibility of rotenone's cytotoxicity in these particular treatment scenarios? 

3.     Fig. 3: Why did we specifically analyze NQO1 when it is well known that cells contain numerous antioxidant-related proteins (e.g., catalase, SOD, etc.) during the pathological process of Parkinson's disease? Furthermore, although the authors indirectly linked protein expression to mRNA expression, the best explanation for NQO1 is based on protein expression. What is the comparison standard (per milligram of total protein) for NQO1 activity analysis? 

4.     Fig. 4: How is the intensity of the Nrf2 signal in the nucleus quantified? Please provide more details. In fact, all nuclei (blue) seem to be equally unblocked by Nrf (green) in all merge fluorescence images. 

5.     Figure 5: To make this result more meaningful, it should be further illustrated with an NRF2 antagonist (e.g., ML385). Rotenone and ECP have the ability to improve p62 performance, so alternative routes may be used instead of Nrf. 

6.     Figure 6: Why is the third bar, representing viability, for the ECP only category the lowest? Writers ought to revise their work with greater care. 

7.     Figure7: Although CC is an inhibitor of AMPK, to prove the importance of AMPK in this mechanism, it is more appropriate to at least provide direct evidence of AMPK phosphorylation changes.

8.     Figure8: How should one choose between high and low dosages? Is it appropriate to administer ECP for 26 days? Exist any published works or early experimental findings that corroborate this?

9.     Fig. 11: The results must be quantified

10.  The effectiveness of BBB penetration is always a top concern for all possible CNS medications. In the Discussion section, the authors mentioned that dieckol in ECP appears to be able to explain this issue. But does dieckol indicate a significant role that ECP can play in the fight against Parkinson's disease? This needs to be explained by the writers. 

11.  There should be scale bars on every image.

Comments on the Quality of English Language

Moderate editing of English language is required

Round 2

Reviewer 2 Report

Comments and Suggestions for Authors

Authors addressed rewiever requests.

Comments on the Quality of English Language

Language is fine. 

Reviewer 3 Report

Comments and Suggestions for Authors

All of my questions have been addressed by the authors. I don't have any more queries.